# Pertechnetate/Perrhenate Surface Complexation on Bamboo Engineered Biochar

**DOI:** 10.3390/ma14030486

**Published:** 2021-01-20

**Authors:** Martin Daňo, Eva Viglašová, Karel Štamberg, Michal Galamboš, Dušan Galanda

**Affiliations:** 1Department of Nuclear Chemistry, Faculty of Nuclear Sciences and Physical Engineering, Czech Technical University in Prague, Břehová 7, 115 19 Prague, Czech Republic; kstamberg@volny.cz; 2Department of Nuclear Chemistry, Faculty of Natural Sciences, Comenius University in Bratislava, Mlynska Dolina, Ilkovičova 6, 842 15 Bratislava, Slovakia; michal.galambos@uniba.sk (M.G.); dusan.galanda@uniba.sk (D.G.)

**Keywords:** engineered biochar, technetium, rhenium, potentiometric titration, sorption, modelling

## Abstract

The work deals with the evaluation of biochar samples prepared from Phyllostachys Viridiglaucescens bamboo. This evaluation consists of the characterization of prepared materials’ structural properties, batch and dynamic sorption experiments, and potentiometric titrations. The batch technique was focused on obtaining basic sorption data of ^99m^TcO_4_^−^ on biochar samples including influence of pH, contact time, and Freundlich isotherm. ReO_4_^−^, which has very similar chemical properties to ^99m^TcO_4_^−^, was used as a carrier in the experiments. Theoretical modeling of titration curves of biochar samples was based on the application of surface complexation models, namely, so called Chemical Equilibrium Model (CEM) and Ion Exchange Model (IExM). In this case it is assumed that there are two types of surface groups, namely, the so-called layer and edge sites. The dynamic experimental data of sorption curves were fitted by a model based on complementary error function erfc(x).

## 1. Introduction

^99^Tc is a long-lived radionuclide, low β energy emitter, and its metastable isomer ^99m^Tc is the most important medical tracer worldwide. The nuclear fuel cycle is the predominant source of ^99^Tc in the environment. The nuclear power plants accidents, the nuclear weapons testing fallout, and the institutional waste (mainly ^99m^Tc radiopharmaceutical products) are much less important sources. The natural Tc occurrence, formed in the Earth’s crust by ^238^U spontaneous fission and neutron-induced ^235^U fission in uranium ores, are negligible. Taking into account the kind of sources and the aqueous chemistry, Tc is released into the environment as pertechnetate (TcO_4_^−^). Its behavior, in the environment attracted much attention of researchers all over the world [1,2,3,4], during the last decades due to its properties, such as long physical half-life, the solubility and mobility of TcO_4_^−^ in aquatic system. Considerable effort has been made to understand the long-term biogeochemical behavior of Tc, its transfer in food chains and the mechanism controlling its mobility in diverse environments [1,5,6]. The pertechnetate ion will not likely be sorbed in significant amounts by soils and suspended bottom sediments of predominantly negative charge but will be highly mobile in soils and waters and thus available for uptake in biota [6].

There are currently several general approaches for TcO_4_^−^ immobilization from waste such as evaporation [7], extraction-chromatographic methods [8], sorption onto carbon-based materials, and their modified forms, i.e., biochar [9], etc. A sorption is an effective approach to TcO_4_^−^ sequestration because it is a simple, convenient, and economical in its own process [10].

The outstanding properties of pyrogenic carbonaceous materials ultimately originate from biomass, i.e., biochar, such as porous structure, enhanced specific surface area, and the content of free functional groups predetermine these materials as a promising and efficient tool for various contaminant immobilization [11,12]. This utilization is valuable from the point of feedstock availability (based on input waste biomass) and cost-effectiveness of sorption material preparation. Processing conditions, production methods, and feedstock characteristics have been shown to affect the final sorption properties of biochar-based sorbents that have been produced in pyrolysis reactors [13]. To be used as tool for contaminants, radionuclides or pollutants immobilization, biochar needs to be engineered, in order to prove its physical–chemical properties, to enhance its suitability and mainly because of the final application. 

Engineered/designer sorbent is commonly used term to indicate supplication-oriented, outcome-based sorbent modification or synthesis. In the recent years, modifications, involving impregnation of mineral sorbents, steam activation, and magnetic modification, etc., have been widely studied [13,14]. The biochar engineering consequences are mainly related to the changes in surface properties, such as surface area increase, surface charge, etc., functional groups and/or the pores volume. The positively charged surface (depending on pH) is typical advantages of non-engineered biochar, resulting in a relatively low sorption capacity of anionic contaminants [15,16]. Based on above mentioned facts, engineered biochars—produced by application-oriented, outcome-based modification, or synthesis—are developed as innovative sorbents with innovative properties for significant improvement of the environmental quality around contaminated regions, and to reduce and eliminate anionic pollutants eco-toxic effects [17,18,19,20,21,22,23,24,25].

In our previous work was shown the fundamental understanding of the changes in the biochar structure as function of various additives is therefore crucial for the implementation of strategies to design/engineer biochar with superior properties, tailored to enhance performance. For this purpose, the biochar composites detailed analysis of structure was described. Prepared and characterized materials were tested for sorption studies of nitrate from aqueous solutions [23]. 

The novelty of this work lies in the description of the interactions of pertechnetate and biochar using potentiometric titrations and the application of the surface-complexation model. Using this model, it is possible to clearly show which types of functional groups are involved in sorbent-adsorbate interactions.

The objectives of this study were to:-enrich and supplement the characterization of prepared materials’ structural properties;-batch sorption investigation influenced by pH, contact time, and equilibrium study;-potentiometric titration and dynamic sorption experiments;-theoretical models’ applications.

All above mentioned experiments and calculations were performed in order to improve the environmental application of prepared engineered biochar for technetium immobilization.

## 2. Materials and Methods

### 2.1. Materials Preparation

Our pyrogenic carbonaceous materials were prepared as follows. As a biomass feedstock for pyrolysis, agricultural bamboo waste from greenway golden bamboo (Phyllostachys viridiglaucescens) was used. First sample consisted of raw bamboo-based biochar labeled as BCS (S means standard). This sample was prepared in the same way as the next two except that no saline solution was used but deionized water (DW) only. Two samples were Mg- and Fe- engineered biochars labelled as BCMg (containing Mg) and BCFe (containing Fe), respectively. The amount of 40 g MgCl_2_·6H_2_O or FeCl_3_·6H_2_O was dissolved in 60 cm^3^ of DW and 150 g of biomass was added to each solution. These suspensions (feedstocks) were stirred for 2 h, then oven-dried at 80 °C for 24 h. Thereafter, dried materials were pyrolyzed at 460 °C for 2 h. After pyrolysis, prepared materials were rinsed with DW and oven dried at 80 °C for 24 h. In order to ensure an oxygen-free environment and uniform heating conditions, nitrogen (N_2_) was used as a flush gas. The samples preparation in detail was well described in our previous work [23]. All chemical solutions were prepared using DW <0.4 μS cm^−1^, which was also used to rinse and wash the samples. All reagents used during experiments were of analytical purity grade.

### 2.2. Materials Characterization

#### 2.2.1. X-ray Fluorescence

The following characterization techniques were carried out for all prepared samples. The X-ray fluorescence (XRF) spectrometer NITON XL3t 900Analyzer with GOLDD Technology (Thermo Scientific, Waltham, MA, USA) was used for qualitative elemental characterization. Before performing the analysis, the NITON was allowed to warm up for a minimum of 15 min. Prior to measurements, the spectrometer was calibrated. The 4 μm thin film (3252 ULTRALANE^®^, Spex SamplePrep, Metuchen, NJ, USA) was stretched at one end of the double-opened ring cup (SC-4331, Premier Lab Supply, Port St. Lucie, FL, USA) and attached with an oversize ring. Samples were sprinkled into these cups. Cups with samples were placed over the detector and measured in “mining” mode.

Except for XRF characterization, all samples were well described in our previous work by field-emission scanning electron microscope (TESCAN MIRA 3, Oxford Instruments, Abingdon, UK) with energy-dispersive X-ray (EDX) detector, FTIR spectroscopy (VERTEX 70, Bruker Optics Inc., Billerica, MA, USA), CHNS elemental analysis (MACRO cube, Elementary Analysensysteme GmbH, Langenselbold, Germany), thermogravimetric analysis (SDTA851e, Mettler Toledo, Columbus, OH, USA), and N_2_; (−196 °C) physical adsorption (NOVA 1200e Surface Area and Pore Size Analyzer, Quantachrom Instruments, Boynton Beach, FL, USA) [23].

#### 2.2.2. Acid-Base Titrations

For the modelling of the titration curves, it is assumed that there are two types of surface functional groups, namely the so-called edge and layer sites. These have different properties, so three types of surface complexation models (SCM) are used to describe edge sites, namely two electrostatic (CCM—Constant Capacitance Model and DLM—Diffusion Double Layer Model) and one without electrostatic correction (CEM—Chemical Equilibrium Model). Processes running on layer sites are described using the IExM—Ion Exchange Model [26]. With respect to basic parameters of the surface sites, acid-based titrations curves were studied. According to Lutzenkirchen [27] reactions taking place on the surface of a biochar can be described by Equations (1)–(3) if CEM and IExM are taken into account.
≡SO^−^ + H^+^ ↔ ≡SOH^0^   K_1_;(1)
≡SOH^0^ + H^+^ ↔ ≡SOH_2_^+^   K_2_(2)
≡XNa + H^+^ ↔ ≡XH + Na^+^   K_ex_(3)
where ≡SO^−^, ≡SOH^0^ and ≡SOH_2_^+^ are symbols for edge sites, X^−^ is the symbol for layer sites. The equilibrium constants K_1_, K_2_ and K_ex_; are given by Equations (4)–(6):(4)$$K_{1}=\frac{\left[{\mathrm{SOH}}^{0}\right]}{\left[{\mathrm{SO}}^{-}\right]\left[H^{+}\right]}$$
(5)$$K_{2}=\frac{\left[{\mathrm{SOH}}_{2}^{+}\right]}{\left[{\mathrm{SOH}}^{0}\right]\left[H^{+}\right]}$$
(6)$$K_{\mathrm{ex}}=\frac{\left[\mathrm{XH}\right]\left[{\mathrm{Na}}^{+}\right]}{\left[\mathrm{XNa}\right]\left[H^{+}\right]}$$

Surface charge density reaction balances for edge sides (7) and layer sides (8) are as follows:(7)$$\sum {\mathrm{SOH}}^{0}=\left[{\mathrm{SOH}}^{0}\right]+\left[{\mathrm{SO}}^{-}\right]+\left[{\mathrm{SOH}}_{2}^{+}\right]$$
(8)$$\sum X=\left[\mathrm{XH}\right]+\left[X^{-}\right]=\left[\mathrm{HX}\right]+\left[\mathrm{XNa}\right]$$

Modeling of titration curves was performed so that in the *i*-th point of the titration curve, the total surface charge density, (Q_cal_)_i_ is equal to the sum of charge density on the edge-sites, (Q_ES_)_i_, and on the layer-sites, (Q_LS_)_i_. Therefore (Q_cal_)_i_ = (Q_ES_)_i_ + (Q_LS_)_i_. Charge density is a function of pH, whereupon the values of (Q_ES_)_i_ (Equation (9)) and (Q_LS_)_i_ (Equation (10)) can be calculated.
(9)$${\left(Q_{\mathrm{ES}}\right)}_{i}=\frac{\sum {\mathrm{SOH}}^{0}\cdot \left(K_{1}\cdot K_{2}\cdot {\left[H^{+}\right]}^{2}+1\right)}{K_{1}\cdot K_{2}\cdot {\left[H^{+}\right]}^{2}+K_{1}\cdot {\left[H^{+}\right]}^{2}+1}$$
(10)$${\left(Q_{\mathrm{LS}}\right)}_{i}=\frac{\sum X\cdot \left[{\mathrm{Na}}^{+}\right]}{\left[{\mathrm{Na}}^{+}\right]+K_{\mathrm{ex}}\cdot \left[H^{+}\right]}$$

The experimental value of the surface charge for *i*-th point of titration curve, (Q_ex_)_i_, can be then calculated according to Equation (11).
(11)$${\left(Q_{\exp}\right)}_{i}=\frac{V_{i}\cdot \left(C_{a,i}-C_{b,i}+{\left[{\mathrm{OH}}^{-}\right]}_{i}-{\left[H^{+}\right]}_{i}\right)}{m_{i}}$$
where V_i_ is the total volume of liquid phase; m_i_ is the mass of solid phase; C_a,i_ and C_b,i_ are bulk concentrations of acid (i.e., NO_3_^−^) and hydroxide (i.e., Na^+^) in liquid phase, respectively. The values of C_a,i_ and C_b,i_ are given by concentrations of acid and base, and by their consumptions in the course of the titration.

If K_1_, K_2_, K_ex_, ∑SOH and ∑X are sought, multi-dimensional non-linear regression procedure can be used. FAMULUS software [28] with Gauss–Newton algorithm was used for calculation. As the criterion of goodness-of-fit WSOS/DF (Weighted Sum of Squares divided by Degrees of Freedom) was chosen (Equation (13)). Its calculation is based on χ^2^-test given by Equation (12).
(12)$$\chi^{2}=\frac{\sum {\left(\mathrm{SSq}\right)}_{i}}{{\mathrm{Sq}}_{i}^{2}}$$
(13)$$\frac{\mathrm{WSOS}}{\mathrm{DF}}=\frac{\chi^{2}}{n_{i}}$$
(14)$$n_{i}=n_{p}-n$$
where (SSq)_i_ is the square of experimental value deviation from calculated one; (Sq)_i_ is the experimental value uncertainty; n_i_ is the number of degrees of freedom calculated by Equation (14); n_p_ is the number of experimental points and n is the number of model parameters sought during the regression procedure.

### 2.3. Batch and Dynamic Sorption Experiments

#### 2.3.1. Speciation Analysis

Because one of the most significant limitations associated with the batch method is that it is not possible to find the difference between the individual species of chemicals by measuring the number of pulses (radioactive measurement), speciation analysis was performed by theoretical and practical means.

Theoretical speciation analysis was performed using dissociation and stability constants from LLNL and ThermoChemie databases. Then, a practical Eh(V) measurement of experimental solutions was carried out in order to verify the speciation of ^99m^TcO_4_^−^ and ReO_4_^−^ (used as carrier). These values were compared with oxidation-reduction potential Eh(V) of Zobell’s solution [29], that was used to the calibration of measuring equipment. Eh(V) was measured with combine pH electrode E16M343 pHC3359-3 (Radiometer Analytical, A Haach Company, Little Island, Ireland). Two Zobell’s standard solutions consist of 0.003 mol·dm^−3^ K_3_Fe(CN)_6_ (p.a., Penta, s.r.o., Prague, Czech Republic), 0.003 mol·dm^−3^ K_4_Fe(CN)_6_ (p.a., Penta, s.r.o., Prague, Czech Republic), and 0.1 mol·dm^−3^ KCl. Eh(V) value of these solutions should be equal to +430 mV for standard hydrogen electrode (SHE).

^99m^TcO_4_^−^ concentrations (c) were obtained by recalculation according to Equation (15) from the activity of the solutions, by random selection in batch experiments. They were measured on a PTW Curiementor 2 ionization chamber (PTW Freiburg, Germany) at iso-factor 527.
(15)$$c=\frac{A}{V\times N_{A}\times \lambda }\;\left(\mathrm{mol}\cdot {\mathrm{dm}}^{-3}\right)$$
where A is activity (Bq; s^−1^), V is the volume (dm^3^), N_A_ is the Avogadro constant (mol^−1^), λ is ^99m^Tc decay constant (s^−1^).

#### 2.3.2. Pertechnetate/Perrhenate Sorption by Batch Technique and Column Porosity Determination

The batch equilibrium study is the one of the most important method of sorption investigation. In this method, 0.02 g of the material was placed into 2 cm^3^ Eppendorf Safe-Lock Tubes. A solution of 0.2 cm^3^ containing sorbate with a respective concentration and pH was added. Tubes were placed in sealed bottles and agitated in a laboratory shaker (IKA Labortechnik KS250 basic shaker). Influence of the pH, the contact time and the amount of carrier are described in more detail below.

The most crucial and important factor in sorption study plays the value of the pH. The aqueous phase pH affects the surface charges, i.e., protonation state of adsorbent functional groups, chemical speciation of Tc(VII) and Re(VII), and diffusion rate of the solute, respectively [30]. Experimental conditions were as follows. Carrier-free stock solutions were prepared with the range of pH 1.0–9.0. The setup of pH was done by HCl (p.a., 35%, Lach-Ner s.r.o., Neratovice, Czech Republic) or NH_3_ (p.a., >24%, Penta, s.r.o, Praha, Czech Republic). Each stock solution was labelled [^99m^Tc]NaTcO_4_ (DRN 4329 Ultra Technekow FM 2.15–43.00 GBq radionuclide generator, Mallinckrodt Medical B.V., Petten, Netherlands), the final volume activity was approx. 1 MBq·cm^−3^. The volume 0.5 cm^3^ of each stock solution was spiked into the 2 cm^3^ PE scintillation vials as a standard. Solid phases and each liquid phase created a set of samples which was shaken in the laboratory shaker at 250 rpm for 24 h at the temperature of 24 ± 1 °C. Then, suspensions were separated by filtration on a glass microfiber filter Whatman GF/C. 0.5 cm^3^ of each filtrate was taken for count rate measurement into PE scintillation vials. Impulse counting was carried out for 100 s in a well-type NaI(Tl) scintillation detector NKG-314 with a single-channel analyzer spectrometric assembly NV-3120 (TESLA VUPJT, Zdiby, Czech Republic).

The respective value of pH, where the highest adsorption percentage (*R*) was reached, served as the environment in equilibrium time determination experiments. Two, carrier-free stock solution (one for BCS, BCMg, one for BCFe) with the specific pH value were prepared (volume activity is approx. equal to 1 MBq·cm^−3^ of [^99m^Tc]NaTcO_4_), and 0.5 cm^3^ of each stock solution was spiked into the 2 cm^3^ PE scintillation vials as a standard. Aqueous and solid phase were mix for 1, 10, 20, 40, 60, 120, and 240 min. Then, suspensions were separated by filtration on a glass microfiber filter Whatman GF/C. 0.5 cm^3^ of each filtrate was taken for count rate measurement into PE scintillation vials. Impulse counting was carried out for 100 s in a well-type NaI(Tl) scintillation detector NKG-314 with a single-channel analyzer spectrometric assembly NV-3120 (TESLA VUPJT, Zdiby, Czech Republic).

The values (pH, contact time) from previous experiments were used for the following, isotherm investigation. Non-isotopic tracer solutions of NH_4_ReO_4_ (≥99%, Sigma-Aldrich, Inc., St. Louis, MO, USA) with concentration of 1 × 10^−7^, 5 × 10^−7^, 1 × 10^−6^, 5 × 10^−6^, 1 × 10^−5^, 5 × 10^−5^, 1 × 10^−4^, 5 × 10^−4^, 1 × 10^−3^, 5 × 10^−3^, and 5 × 10^−2^ mol·dm^−3^ prepared and labelled with [^99m^Tc]NaTcO_4_. The final volume activity was approx. 1 MBq·cm^−3^. Suspensions were mix for the specific time. Then, suspensions were separated by filtration on a glass microfiber filter Whatman GF/C. 0.5 cm^3^ of each filtrate was taken for count rate measurement into PE scintillation vials. Impulse counting was carried out for 100 s in a well-type NaI(Tl) scintillation detector NKG-314 with a single-channel analyzer spectrometric assembly NV-3120 (TESLA VUPJT, Zdiby, Czech Republic).

The mathematical processing of radioactivity measurements is as follows. The background count rate (n_b_) is the number of background counts (N_b_) divided by measuring time (t_b_).
(16)$$n_{b}=\frac{N_{b}}{t_{b}}\;\left(s^{-1}\right)$$

The error of the background count measurement (σ(N_b_)) can be calculated according to Equation (17).
(17)$$\sigma \left(N_{b}\right)=\sqrt{N_{b}}\;\left(\mathrm{pulse}\right)$$

Then, using the law of error propagation, it is possible to calculate the error of the background count rate (Equation (18)).
(18)$$\sigma \left(n_{b}\right)=\sqrt{\frac{n_{b}}{t_{b}}}=\frac{\sigma \left(N_{b}\right)}{t_{b}}\;\left(s^{-1}\right)$$

Similarly, other values are obtained, sample + background count rate (n_s+b_), its error (σ(n_s+b_)), standard + background count rate (n_st+b_), and its error ((σ(n_st+b_)) as follows.
(19)$$n_{s+b}=\frac{N_{s+b}}{t_{s+b}}\;\left(s^{-1}\right)$$
(20)$$\sigma \left(n_{s+b}\right)=\sqrt{\frac{n_{s+b}}{t_{s+b}}}=\frac{\sigma \left(N_{s+b}\right)}{t_{s+b}}\;\left(s^{-1}\right)$$
(21)$$n_{\mathrm{st}+b}=\frac{N_{\mathrm{st}+b}}{t_{\mathrm{st}+b}}\;\left(s^{-1}\right)$$
(22)$$\sigma \left(n_{\mathrm{st}+b}\right)=\sqrt{\frac{n_{\mathrm{st}+b}}{t_{\mathrm{st}+b}}}=\frac{\sigma \left(N_{\mathrm{st}+b}\right)}{t_{\mathrm{st}+b}}\;\left(s^{-1}\right)$$

The net sample (n_s_) and net standard (n_st_) count rates and their errors σ(n_s_) and σ(n_st_) are then:(23)$$n_{s}=n_{s+b}-n_{b}\;\left(s^{-1}\right)$$
(24)$$n_{\mathrm{st}}=n_{\mathrm{st}+b}-n_{b}\;\left(s^{-1}\right)$$
(25)$$\sigma \left(n_{s}\right)=\sqrt{{\left(\sigma \left(n_{s+b}\right)\right)}^{2}+{\left(-\sigma \left(n_{b}\right)\right)}^{2}}\;\left(s^{-1}\right)$$
(26)$$\sigma \left(n_{\mathrm{st}}\right)=\sqrt{{\left(\sigma \left(n_{\mathrm{st}+b}\right)\right)}^{2}+{\left(-\sigma \left(n_{b}\right)\right)}^{2}}\;\left(s^{-1}\right)$$

The number of pulses of a radionuclide decreases over time. The half-life of ^99m^Tc is quite short, equal to 6.0067 h. Therefore, the sample count rates (n_s_) (Equation (23)) need to be corrected (n_cor_) (Equation (27)) by the time (t_cor_) that elapses between the standard measurement and the measurement of each sample.
(27)$$n_{\mathrm{cor}}=\frac{n_{s}}{e^{-{\lambda t}_{\mathrm{cor}}}}\;\left(s^{-1}\right)$$
(28)$$\sigma \left(n_{\mathrm{cor}}\right)=\sigma \left(n_{s}\right)\times e^{-{\lambda t}_{\mathrm{cor}}}\;\left(s^{-1}\right)$$
where λ is the ^99m^Tc decay constant (3.20543 × 10^−5^ s^−1^).

The portion of adsorbed ^99m^TcO_4_^−^ is given by the difference of its concentrations or its count rates in the aqueous phase before and after mixing. This portion described by sorption percentage R.
(29)$$R=100-\left(\frac{n_{\mathrm{cor}}}{n_{\mathrm{st}}}\right)\times 100\;\left(\%\right)$$
(30)$$\sigma \left(R\right)=\sqrt{{\left(-100\times \frac{1}{n_{\mathrm{st}}}\times \sigma \left(n_{\mathrm{cor}}\right)\right)}^{2}+{\left(100\times \frac{n_{\mathrm{cor}}}{n_{\mathrm{st}}^{2}}\times \sigma \left(n_{\mathrm{st}}\right)\right)}^{2}}\;\left(\%\right)$$

The mass distribution ratio (D_g_) represents a sorbent relationship to the adsorbate. It is a linear dependency of equilibrium concentration of adsorbed substance to its equilibrium concentration in the solution.
(31)$$D_{g}=\mathrm{BF}\times \frac{n_{\mathrm{st}}-n_{\mathrm{cor}}}{n_{\mathrm{cor}}}\;\left({\mathrm{cm}}^{3}\cdot g^{-1}\right)$$
(32)$$\sigma \left(D_{g}\right)=\sqrt{{\left(\mathrm{BF}\times \frac{1}{n_{\mathrm{cor}}}\times \sigma \left(n_{\mathrm{st}}\right)\right)}^{2}+{\left(-\mathrm{BF}\times \frac{n_{\mathrm{st}}}{n_{\mathrm{cor}}^{2}}\times \sigma \left(n_{s}\right)\right)}^{2}}\;\left({\mathrm{cm}}^{3}{\;g}^{-1}\right)$$
where BF = V/m, m is the solid phase mass (g), and V is the liquid phase volume of liquid phase (cm^3^).

The D_g_ values range normally from 10^0^ to 10^4^ cm^3^·g^−1^. Therefore, it is useful to determine whether the values are within this range. That means making an estimating the minimum (D_g,min_) and maximum (D_g,max_) value. If the time of sample + background and standard + background count measurements is the same (t = t_s+b_ = t_st+b_), Equation (32) can be modified to the following form.
(33)$$\sigma \left(D_{g}\right)=\frac{1}{n_{s+b}-n_{b}}\times \mathrm{BF}\times \sqrt{\frac{n_{\mathrm{st}}}{t}+{\left(\frac{n_{\mathrm{st}+b}-n_{b}}{n_{s+b}-n_{b}}\right)}^{2}\times \frac{n_{s+b}}{t}+{\left(\frac{n_{\mathrm{st}+b}-n_{b}}{n_{s+b}-n_{b}}\right)}^{2}\times \frac{n_{b}}{t_{b}}}\;\left({\mathrm{cm}}^{3}\cdot g^{-1}\right)$$

If we assume the Gaussian distribution of pulses, the number of pulses is high enough, and at the same time the sample count rates are close to standard count rates, then assuming that n_st+b_ ⨠ n_b_ it is possible to write the limit of σ(D_g_) as follows:(34)$$\underset{n_{s+b}\to n_{\mathrm{st}+b}}{\lim}\sigma \left(D_{g}\right)=\frac{1}{n_{\mathrm{st}+b}-n_{b}}\times \mathrm{BF}\times \sqrt{\frac{2}{n_{\mathrm{st}+b}\times t}}\approx \mathrm{BF}\times \sqrt{\frac{2}{n_{\mathrm{st}+b}\times t}}$$

The minimum weight distribution ratio (D_g,min_) was expressed by selecting the 99.7% confidence interval in which the mean D_g_ values lie, i.e., where coverage factor *k* = 3.
(35)$$D_{g,\min}=3\times \sigma \left(D_{g}\right)=3\times \mathrm{BF}\times \sqrt{\frac{2}{n_{\mathrm{st}+b}\times t}}\;\left({\mathrm{cm}}^{3}\cdot g^{-1}\right)$$

If the value of the minimum count rate that we can prove (n_s+b_) is expressed as follows.
(36)$$n_{s+b}=n_{b}+3\times \sigma \left(D_{g}\right)\;\left(s^{-1}\right)$$

Then, it holds that the maximum distribution ratio value (n_st+b_ ⨠ n_b_) can be written as follows.
(37)$$D_{g,\max}=\mathrm{BF}\times \frac{n_{\mathrm{st}}-n_{b}-3\times \sqrt{\frac{n_{b}}{t_{b}}}}{3\times \sqrt{\frac{n_{b}}{t_{b}}}}\approx \mathrm{BF}\times \frac{n_{\mathrm{st}}}{3}\times \sqrt{\frac{t_{b}}{n_{s+b}}}\;\left({\mathrm{cm}}^{3}\cdot g^{-1}\right)$$

Equilibrium concentration (c_eq_) indicates what part of carrier analytical concentration (c) remained in the liquid phase after adsorption process.
(38)$$c_{\mathrm{eq}}=c\times \frac{n_{\mathrm{cor}}}{n_{\mathrm{st}}}\;\left(\mathrm{mol}\cdot {\mathrm{dm}}^{-3}\right)$$
(39)$$\sigma \left(c_{\mathrm{eq}}\right)=\sqrt{{\left(\frac{c}{n_{\mathrm{st}}}\times \sigma \left(n_{\mathrm{cor}}\right)\right)}^{2}+{\left(-c\times \frac{n_{\mathrm{cor}}}{n_{\mathrm{st}}^{2}}\right)}^{2}}\;\left(\mathrm{mol}\cdot {\mathrm{dm}}^{-3}\right)$$

For the isotherms modeling it is necessary to know what amount of the sorbate is sorbed per gram of a sample. This expresses the capacity Q.
(40)$$Q=D_{g}\times c_{\mathrm{eq}}\;\left(\mathrm{mmol}\cdot g^{-1}\right)$$
(41)$$\sigma \left(Q\right)=\sqrt{{\left(c_{\mathrm{eq}}\times \sigma \left(D_{g}\right)\right)}^{2}+{\left(D_{g}\times \sigma \left(c_{\mathrm{eq}}\right)\right)}^{2}}\;\left(\mathrm{mmol}\cdot g^{-1}\right)$$

In this work, the sorption is described by the Freundlich isotherm (q_F_) (Equation (42)). A set of carrier solutions of NH_4_ReO_4_ (≥99%, Sigma-Alrich, Saint Louis, MO, USA) with the analytical concentrations of (c_anal_) 1 × 10^−7^, 5 × 10^−7^, 1 × 10^−6^, 5 × 10^−6^, 1 × 10^−5^, 5 × 10^−5^, 1 × 10^−4^, 5 × 10^−4^, 1 × 10^−3^, 5 × 10^−3^, and 5 × 10^−2^ mol·dm^−3^ with the corresponding pH were prepared. Each carrier solution was labelled with [^99m^Tc]NaTcO_4_ (DRN 4329 Ultra Technekow FM 2.15–43.00 GBq radionuclide generator, Mallinckrodt Medical B.V., Petten, Netherlands), with the final volume activity of ∼1 MBq·cm^−3^. The volume of generator solution did not exceed 100 μL in either case. The standard volume 0.5 cm^3^ was taken from each concentration for radioactivity measurement. The amount 20 mg of each sample was contacted with 2 cm^3^ each prepared concentration in 2 cm^3^ safe-lock tubes (Eppendorf, Hamburg, Germany). Then, tubes were shaken in a shake at 250 rpm. After filtration on a Whatman GF/C glass microfiber filter (Whatman plc., Maidstone, UK), 0.5 cm^3^ was taken from each filtrate for impulse counting (100 s) in a well-type NaI(Tl) scintillation detector NKG-314 with a single-channel analyzer spectrometric assembly NV-3120 (TESLA VUPJT, Zdiby, Czech Republic).
(42)$$q_{F}=k_{F}\times c_{\mathrm{eq}}^{n_{F}}\;\left(\mathrm{mmol}\cdot g^{-1}\right)$$
where k_F_ is the Freundlich constant, n_F_ is the regression constant. If n_F_ = 1, then k_F_ = D_g_. In all batch experiments the lower detection limit lied below 1%. Graphs were plotted in Origin 9.5.

#### 2.3.3. Pertechnetate/Perrhenate Sorption by Dynamic Technique

Breakthrough curve is the result of the dynamic experiment under forced flow condition on a fixed bed in a column. The position and shape of this curve is considerably influenced by sorption capacity, selectivity, release, and transfer of heat as well as the sorption rate, inlet concentration, flow rate, and by the limited experimental time due to the short ^99m^Tc half-life. Therefore, suitable conditions have been sought experimentally and are as follows. In this work, the breakthrough curve was measured as a dependence of relative count rate (*n_rel_*)*_theor_*, on the rate of flow expressed as the number of the sorbent bed volume. Here (*n_rel_*)*_theor_* is the theoretical, relative count rate which corresponds to the ReO_4_^−^ concentration ratio at the given point of breakthrough curve (*c^∗^*) to the inlet sorption solution concentration (*c_0_*). 

The 1 cm^3^ empty Rezorian™ tube kit with PE first (Supelco Inc, Bellefonte, PA, USA) was used as a column. The column’s internal dimensions are 2.1 cm × 0.45 cm, i.e., one bed volume (BV) is equal to 1.336 cm^3^. The composition of sorption solutions and bed weights (*m*) are in Table 1. The sorption solutions consisted of carrier NH_4_ReO_4_ (≥99%, Sigma-Aldrich, Saint Louis, MO, USA), pH was adjusted with HCl (p.a., 35%, Lach-Ner Ltd., Neratovice, Czech Republic). Each sorption solution was labelled by [^99m^Tc]NaTcO_4_ (DRN 4329 Ultra Technekow FM 2.15–43.00 GBq radionuclide generator, Mallinckrodt Medical B.V., Petten, The Netherlands). The resulting volume activity of each sorption solution was approximately 1 MBq·cm^−3^. 0.5 cm^3^ of each sorption solution was saved into the 2 cm^3^ PE scintillation vials as a standard.

Columns were packed and slightly pressed manually. Each column was conditioned in a slow, upward flow direction at the room temperature with approximately 20 BV of DW. All columns rested for at least 8 h. Then, created bubbles in columns were eluted by approximately 10 BV of DW, lower 2-way valve was closed, water above the column was removed, hoses were drained and dried. The column remained water-flooded (1 BV) prior to sorption study. Then, the inlet hoses were rinsed (into the waste bottle) with approximately 10 cm^3^ of labeled sorption solution. Then, flow was stopped, lower 2-way was turned to the flow-through position with the column, flow was switched on. The fractions collection began with the first drop at the house output. The linear flow rate was set to 0.2 cm·min^−1^ (4.5 BV·h^−1^), flow was secured by the peristaltic pump PCD22 (Dávkovací čerpadla—Kouřil, Kyjov, Czech Republic) and the fraction collector 2210 (Bio-Rad spol s.r.o., Prague, Czech Republic) was used to collect the eluates into PE scintillation vials every 6 min. 0.5 cm^3^ of each fraction was measured for 100 s in a well-type NaI(Tl) scintillation detector NKG-314 with a single-channel analyzer spectrometric assembly NV-3120 (all Tesla, Zdiby, Czechoslovakia). After ^99m^Tc decay, equilibrium pH was measured by pH meter PHM2200 equipped with a combined pH electrode XC161-9 (Radiometer Analytical, Washington, WA, USA). Graphs were plotted in Origin 9.5.

The experimental points were fitted by transport model which is based on complementary error function (erfc(*x*)). This function was obtained by analytical solution of one-dimensional advective dispersion Equation (43) for steady seepage flow [31].
(43)$$\varepsilon \times D_{l}\times \frac{\partial^{2}c}{\partial x^{2}}-\varepsilon \times u\times \frac{\partial c}{\partial x}=\varepsilon \times \frac{\partial c}{\partial t}+\xi \times \frac{\partial q}{\partial t}-\varepsilon \times \lambda \times C-\varepsilon \times \lambda \times q$$
where ε is the porosity (cm^3^·cm^−3^), D_l_ is the dispersion coefficient (cm^2^·s^−1^), c is the carrier concentration (mmol·cm^−3^), x is the distance to which sorbate travels in the bed (cm), u is the linear velocity of the liquid phase (cm·s^−1^), ξ = m/BV is the bulk density of the bed of solid phase (g·cm^−3^), q is the amount of sorbed carrier (mmol·g^−1^), t is the time (s), λ is the decay constant of ^99m^Tc (s^−1^). It is obvious that the retardation coefficient is a function of the first derivative of the relation for the equilibrium isotherm at point *c*. If the Freundlich Equation (42) is considered, then the first derivative is given by Equation (44):(44)$$f\prime \prime =\frac{\partial q}{\partial c}=n_{F}\times k_{F}\times {\left({\left(n_{\mathrm{rel}}\right)}_{\mathrm{theor}}\times c_{0}\right)}^{\left(n_{F}-1\right)}$$

The sorption dynamics process depends on the initial boundary conditions. Therefore, the correct formulation of these conditions is an integral part of the model’s mathematical formulation. Assuming that the sorption process is isothermal, the mobile phase is incompressible, flows in one direction and has a statistical character, the sorbate concentration is so small that local changes in fluid density are negligible. The initial and boundary conditions are then [31]:(45)$$t=0,\;0\;\leq x\leq x_{0},\;c=0\;\mathrm{at}\;x=0\;\mathrm{and}\;t=0,\;c_{t}\;=\;c_{0}\;\mathrm{at}\;x\;=\;L$$

Then, the Equation (44) can be written as follows [32]:(46)$$c=\frac{c_{0}}{2}\times \left[\mathrm{erfc}\left(\frac{x-\frac{u}{R}\times t}{\sqrt[{2}]{\frac{D_{l}\times t}{R}}}\right)+e^{\frac{u\times x}{D_{d}}\times \mathrm{erfc}\left(\frac{x-\frac{u}{R}\times t}{\sqrt[{2}]{\frac{D_{l}\times t}{R}}}\right)}\right]$$

Neglecting the exponential term and modification for count rate [33]:(47)$${\left(n_{\mathrm{rel}}\right)}_{\mathrm{theor}}=\frac{{\left(n_{o}\right)}_{\mathrm{theor}}}{n_{\mathrm{st}}}=0.5\cdot \mathrm{erfc}\left(\frac{{\left(R_{s}\right)}_{\mathrm{theor}}-n_{\mathrm{pv}}}{2\cdot \sqrt{\frac{{\left(R_{s}\right)}_{\mathrm{theor}}\times n_{\mathrm{pv}}}{P_{e}}}}\right)$$

Because it holds that erfc(*-x*) = 2 − erfc(*x*), Equation (46) can be written as follows:(48)$${\left(n_{\mathrm{rel}}\right)}_{\mathrm{theor}}=\frac{{\left(n_{o}\right)}_{\mathrm{theor}}}{n_{\mathrm{st}}}=1-\left[0.5\times \mathrm{erfc}\left(-\frac{{\left(R_{s}\right)}_{\mathrm{theor}}-n_{\mathrm{pv}}}{2\cdot \sqrt{\frac{{\left(R_{s}\right)}_{\mathrm{theor}}\times n_{\mathrm{pv}}}{P_{e}}}}\right)\right]$$
where (n_0_)_theor_ is the theoretical value of corrected count rate of the liquid phase at the outlet of the column (s^−1^), n_st_ is the standard’s count rate (s^−1^), n_pv_ (= u × t/L) is the experimental count of bed volumes (BV), L is the bed length, t is the time of the sorption experiment, P_e_ (= u × L/D_d_) is the Peclet number, and D_d_ is the dispersion coefficient (cm^2^·min^−1^).

According to Equation (46), the calculation is possible if (n_rel_)_theor_ ≤ 0.5, i.e., (R_s_)_theor_ − n_pv_ ≥ 0. If (n_rel_)_theor_ > 0.5, i.e., (R_s_)_theor_ − n_pv_ < 0, Equation (47) must be used.

Porosity ε (cm^3^·cm^−3^), sample density ρ_s_ (g·cm^−3^), bulk density of the sample in the column ξ (g·cm^−3^), and linear flow rate u (cm·min^−1^) need to be determined experimentally because they enter the computational model. A mass of dry, empty pycnometer (m_e_) and the same one filled with deionized water (m_f_) was weighed. The pycnometer was then thoroughly dried, and a dry sample (m_s_) was weighed into it. Subsequently, the sample in the pycnometer was wetted so that the volume of deionized water was approximately half the total volume of the pycnometer. Air from the samples’ pores was evacuated under reduced pressure for 10–15 min. The pycnometer was then filled to the throat, sealed, dried, and weighed (m_s+w_). From the obtained masses it is possible to calculate the sample volume V_s_ and the density ρ_s_ of the sample then according to the relations:(49)$$V_{s}=\left(m_{f}-m_{e}\right)-\left[m_{s+w}-\left(m_{e}+m_{s}\right)\right]\;\left({\mathrm{cm}}^{3}\right)$$
(50)$$\rho_{s}=\frac{m_{s}}{V_{s}}\;\left(g\cdot {\mathrm{cm}}^{3}\right)$$

The quantities written above were used to calculate the net volume (V_n_).
(51)$$V_{n}=\frac{m}{\rho }\;\left({\mathrm{cm}}^{3}\right)$$
where m (g) is the dry mass of the sample in the column (see Table 1). While the bed volume, BV, is equal to 1.336 cm^3^, then the porosity ε and the bulk density of the sample in the column ξ can be calculated as follows:(52)$$\varepsilon =\frac{\mathrm{BV}-V_{n}}{\mathrm{BV}}\;\left({\mathrm{cm}}^{3}\cdot {\mathrm{cm}}^{-3}\right)$$
(53)$$\xi =\frac{m}{\mathrm{BV}}\;\left(g\cdot {\mathrm{cm}}^{-3}\right)$$

The samples were measured three times, the resulting values of ρ_s_, ε and ξ were averaged. If the porosity ε is known, the average solution volume flowing through the column per minute V_c_ (cm^3^·min^−1^) was determined by weighing the fractions, the linear flow rate of the solution through the column u (cm·min^−1^) can be calculated by means of Equations (53) and (54):(54)$$S_{c}=\pi \cdot r^{2}\cdot \varepsilon \cdot \left({\mathrm{cm}}^{2}\cdot {\mathrm{cm}}^{3}\cdot {\mathrm{cm}}^{-3}\right)$$
(55)$$u=\frac{V_{c}}{S_{c}}\;\left(\mathrm{cm}\cdot \min^{-1}\right)$$
where S_c_ is the column free cross section.

## 3. Results and Discussion

### 3.1. Material Characterization

#### 3.1.1. XRF

XRF spectroscopy provided identifiable elements between Al and W. The results are shown in Table 2. The background represents Fe, Ni, and W. All identified elements are macro- and microbiogenic in nature. This means that they have been ingested from the outside environment and the plant incorporates them into its metabolism. Si is not essential element, but it seems could be beneficial for vegetation according to [34].

Although the magnesium and aluminum peaks have been identified, their intensities are very small due to the low K energy levels. Therefore, as we reported before in [23], EDX was used to identify elements heavier than Be. The measurement was performed on two randomly selected areas within one particle or on two different BC particles of the same sample. The majority component consists of C and O for the BCS sample, for BCMg and BCFe samples the majority corresponds to the compound with which they were impregnated. The weight percentages of the elements obtained by this method vary depending on the place where the measurement took place. This is interpreted by the BCMg sample, which shows a very large difference in the weight percentages of Mg. The difference is up to 39.6%. Identified elements by EDX are shown in Table 3.

#### 3.1.2. Acid-Base Titration

The result of an acid-base titration is shown in Figure 1. This is dependence of the total surface charge on pH for the experimental titration points and for the model. The important result is the pH at which the charge of the surface is equal to zero (pH_PZC_). pH_PZC_ value for BCS is equal to 6.7. This value of BC prepared from *Phyllostachys viridiglaucescens* bamboo is comparable to the one prepared by Hu [5]. The sample pyrolyzed at 300 °C reached a pH_PZC_ of 6.82.

The application of nonlinear regression procedure during fitting of experimental data requires that estimates of the sought variables be entered. These were as follows, log(K_1_) = 15, log(K_2_) = 10, log(K_ex_) = 5, [SOH]_tot_ = [S]_tot_ = 0.1 mol·kg^−1^ The result of the calculation is shown in Table 4.

Currently, there is no analytical technique that can distinguish functional groups on edge sites and layer sites. The model of titration curves is based on the existence of two functional groups. The results of the titrations indicate that there are two groups on which acids and bases act. This means that they are dissociable and have a pH-dependent surface charge. This means that they are considered as edge sites. Based on the WSOS/DF fit criterion, it can be concluded that the model describes the properties of functional groups very well. This supports the assumption of the applicability of edge and layer sites of the given sorbent sample. From the point of view of sorption of anionic species of Tc(VII) and Re(VII), the dependence SOH_2_^+^ on pH in Figure 1b is important. It is evident that their sorption has favorable conditions in almost the entire pH range, especially in the acidic one.

### 3.2. Batch Adsorption Experiments

#### 3.2.1. Speciation Study

What chemical form is present in the solution is very important information, even in connection with batch experiments. Because one of the most significant limitations associated with the batch method is that it is not possible to find the difference between the individual species of chemicals by measuring radioactivity.

The considerations and results of the speciation analysis are as follows. From a radiochemical point of view, the radioactive isotope ^99m^Tc(VII) is present in this system as a tracer (radioindicator) and Re(VII) as a non-isotopic carrier with close chemical properties. Dissociation constants for ionic strength I = 0 mol·dm^−3^ were found only for HTcO_4_, KTcO_4_, and NaTcO_4_ in the HATCHES database (currently replaced by the ThermoChimie database) and in the LLNL database. There are no database data for compounds of ReO_4_^−^ yet.
HTcO_4_ ↔ H^+^ + TcO_4_^−^     log*k*_1_ = 5.94 (5.96 LLNL)(56)
KTcO_4_ ↔ K^+^ + TcO_4_^−^     log*k*_2_ = −2.29 (−2.27 LLNL)(57)
NaTcO_4_ ↔ Na^+^ + TcO_4_^−^     log*k*_3_ = 1.56 (1.52 LLNL)(58)
NH_4_TcO_4_ ↔ 5e^−^ + 4H^+^ + NO_3_^−^ + TcO^2+^     log*k*_4_ = −86.56 (ThermoChemie)(59)

From the value of log*k*_4_ and from the reactions themselves it is clear that in equilibrium the reaction (54) is significantly shifted to the left, and NO_3_^−^ is a good oxidizing agent Tc(IV) → Tc(VII), in an acidic environment to form TcO_4_^−^ and NH_4_^+^. Considering the chemical similarity of Tc and Re, it can be assumed that Equations (51)–(54), including the values of equilibrium constants, can be also apply for ReO_4_^−^. It follows from Equation (51) that HTcO_4_ and HReO_4_ are strong acids, as evidenced by the very low value of molar fractions F2 (degree of dissociation) in Table 5.

Since the dissociation constant of NH_4_TcO_4_ is not known, the log*k*_2_ constant for KTcO_4_ was used. Therefore, it is possible to assume the analogy of the ^99m^TcO_4_^−^ (ReO_4_^−^) reactivity with the reactivity (sorption) on a strongly acidic cation exchange resin, i.e., with the functional group HSO_3_^−^, whose reactivities are in the order Na^+^<NH_4_^+^<K^+^. Therefore, by choosing log*k*_2_, the proportion of free ^99m^TcO_4_^−^ (ReO_4_^−^) in the solution was relatively disadvantaged—see F1 in Table 5.

For the calculation of the representation of the forms ^99m^TcO_4_^−^, resp. ReO_4_^−^ it is necessary to derive the following balance equations:(60)$$\left[{\mathrm{TcO}}_{4}^{-}\right]=\frac{\sum \left({\mathrm{TcO}}_{4}^{-}\right)}{\left(1+\frac{\left[H^{+}\right]}{k_{1}}+\frac{\left[K^{+}\right]}{k_{2}}\right)}$$
(61)$$\left[{\mathrm{HTcO}}_{4}\right]=\frac{\frac{\left[H^{+}\right]\left[{\mathrm{TcO}}_{4}^{-}\right]}{k_{1}}}{\sum \left({\mathrm{TcO}}_{4}^{-}\right)}$$
(62)$$\left[{\mathrm{KTcO}}_{4}\right]=\frac{\frac{\left[K^{+}\right]\left[{\mathrm{TcO}}_{4}^{-}\right]}{k_{2}}}{\sum \left({\mathrm{TcO}}_{4}^{-}\right)}$$
(63)$$F1=\frac{\left[{\mathrm{TcO}}_{4}^{-}\right]}{\sum \left({\mathrm{TcO}}_{4}^{-}\right)}$$
(64)$$F2=\frac{\left[{\mathrm{HTcO}}_{4}\right]}{\sum \left({\mathrm{TcO}}_{4}^{-}\right)}$$
(65)$$F3=\frac{\left[{\mathrm{KTcO}}_{4}\right]}{\sum \left({\mathrm{TcO}}_{4}^{-}\right)}$$
(66)$$\left[{\mathrm{ReO}}_{4}^{-}\right]=\frac{\sum \left({\mathrm{ReO}}_{4}^{-}\right)}{\left(1+\frac{\left[H^{+}\right]}{k_{1}}+\frac{\left[K^{+}\right]}{k_{2}}\right)}$$
(67)$$\left[{\mathrm{HReO}}_{4}\right]=\frac{\frac{\left[H^{+}\right]\left[{\mathrm{ReO}}_{4}^{-}\right]}{k_{1}}}{\sum \left({\mathrm{ReO}}_{4}^{-}\right)}$$
(68)$$\left[{\mathrm{KReO}}_{4}\right]=\frac{\frac{\left[K^{+}\right]\left[{\mathrm{ReO}}_{4}^{-}\right]}{k_{2}}}{\sum \left({\mathrm{ReO}}_{4}^{-}\right)}$$
(69)$$F1=\frac{\left[{\mathrm{ReO}}_{4}^{-}\right]}{\sum \left({\mathrm{ReO}}_{4}^{-}\right)}$$
(70)$$F2=\frac{\left[{\mathrm{HReO}}_{4}\right]}{\sum \left({\mathrm{ReO}}_{4}^{-}\right)}$$
(71)$$F3=\frac{\left[{\mathrm{KReO}}_{4}\right]}{\sum \left({\mathrm{ReO}}_{4}^{-}\right)}$$

The calculated mole fraction values F1, F2, and F3, given by Equations (64)–(66), for different KReO_4_ input concentrations and two pH intervals are in Table 5. The results were obtained in FAMULUS software with the programme PTC_2.fm [26]. The pH of 0–10 was chosen in order to verify the effect even in the alkaline range. The effect in the basic range is practically zero, only the values of F2 have a few orders of magnitude lower value at pH 10, which is logical but insignificant. Values of F1, i.e., representation of free [ReO_4_^−^] and [^99m^TcO_4_^−^] are mainly influenced by the concentration of the KReO_4_ carrier greater than 10^−3^ mol·dm^−3^, as evidenced by F3 values. In all the examples considered, the proportion of free acid [HReO_4_] and [HTcO_4_] presented by F2 is negligible. However, it should be noted that it is very likely that the reduced concentration of [ReO_4_^−^] and [TcO_4_^−^] does not mean a reduction in the sorption efficiency of this anionic form. From the analogy of sorption, e.g., U(VI) from solutions containing SO_4_^2−^, where both anionic and cationic forms of U(VI) exist, it is known that sorption on both anion exchangers and cation exchangers takes place practically at 99% [35]. This is due to the fact that during sorption, decomplexation or complexation occurs, and thus a species is formed which is sorbed.

It is essential that the redox potential Eh(V) of the system be such that there is practically only one form of Tc and Re, i.e., ^99m^TcO_4_^−^ and ReO_4_^−^, respectively. According to the Takeno’s atlas of Eh(V)-pH diagrams [34] at pH 2, Eh(V) should be greater than 500 mV, and at pH 4 greater then 300 mV so that only TcO_4_^−^ is present in the system. This should apply to both Tc(VII) and Re(VII). Practical verification of Eh(V) measurements is in Table 6 (Eh(V) was measured with combine pH electrode E16M343 pHC3359-3 (Radiometer Analytical, A Haach Company, Little Island, Ireland—see Section 2.2.1).

Conversion of Eh(V) values in Table 6 to SHE one, it can be estimated using the measured value of Eh(V) of the Zobell solution which is ~99 mV. Evidently, the absolute difference of this Eh(mV) as compared with SHE data, is then ~530 mV. Since the Eh(mV) of the samples was ~270 mV, after adding the difference of 530 mV, the resulting estimate of SHE Eh(mV) of the samples is equal to ~800 mV. This is a sufficiently high value on the base of which can be claimed that only ReO_4_^−^ and ^99m^TcO_4_^−^ are present in the sorption experiments.

Concentrations of ^99m^TcO_4_^−^ calculated according to Equation (15) were in the rage of 10^−12^–10^−10^ mol·dm^−3^. It should be noted that the tracer concentrations of ΣTc may be underestimated by several orders of magnitude. The reason is the decay of ^99m^Tc to ^99^Tc, which behaves in the same way as a metastable isotope but is not detectable by the detector. However, its concentration is small, and it follows from the above that that the presence of ^99^Tc after the conversion of ^99m^Tc does not and cannot affect the speciation and the experimental results themselves.

#### 3.2.2. Dependence of Sorption Percentage R on pH

The results of the pH measurement are shown in Figure 2. Black dots correspond to pH values of solutions before sorption, red points to equilibrium pH values after the sorption. These experiments were performed without a carrier. For the BCS sample, the pH_eq_ values are kept below 6 (Figure 2a). From this it can be concluded that the excess of H^+^ caused a decrease in R, which increases up to pH 4, where it reaches a maximum. Then the number of active sorption sites decreases with increasing pH.

A surprising result of the dependence of R on pH was obtained for sample BCMg, which was non-thermally pretreated with MgCl_2_ solution. In the case that MgCl_2_ is released from the surface of BCMg and dissolved in the aqueous phase, it hydrolyzes to various hydrates of MgCl_2_·nH_2_O forming slightly acidic solution (pH~6) [36]. Therefore, pH_eq_ was expected to follow a similar trend as the BCS sample. It can be seen (Figure 2b) that all pH_eq_ lie around 10. According to Takeno’s Eh-pH diagrams [36], only the Mg^2+^ and of Cl^−^ are present in the range of pH 0–11.9. Thus, it is not entirely clear what causes such a loss of H^+^ ions regardless of their concentration in aqueous phase. This may be due to various reactions of the polyatomic and/or hydrated molecules in solution with sample’s functional groups or other interactions. In addition, at pH > 7, the number of active sites, and thus the R values is reduced by only a few %. This means that this sample is usable for sorption of ^99m^TcO_4_^−^ in the pH range from 1 to 10, but desorption of ^99m^TcO_4_^−^ (and reusability of the BCMg) will not be possible under these conditions.

The dependence of R on pH for sample BCFe is shown in Figure 2c. Regardless of the initial pH, the pH_eq_ values are between 1 and 2. FeCl_3_, which contains a sample of BCFe, has a strong tendency to hydrolyze and to form various mononuclear and polynuclear complexes of Fe_x_O_i_(OH)_v_Cl_u_ [37,38]. The FeCl_3_ is a salt of a weak base and a strong acid, therefore its solutions have a pH < 7. However, the same was true for the sample BCMg, such acidification of the solution is negligible due to the results. Even with this sample, TcO_4_^−^ sorption is possible in the entire investigated pH range. Verification of the R accuracy was performed using the maximum and minimum D_g_. As can be seen from Table 7, all experimental D_g_ values lie between the minimum and maximum D_g_.

#### 3.2.3. Determination of the Sorption Equilibrium Time

The achievement of sorption equilibrium is relatively slow for all investigated BC samples. In Figure 3 can be seen that BCMg and BCFe reached it almost 100% approximately in 240 min, BCS reached only 78% at the same time. This is the direct cause of sample pretreatment which caused increase in specific surface area, i.e., for BCMg: 75 m^2^·g^−1^, for BCFe: 7 m^2^·g^−1^, while the value for the BCS was only 3 m^2^·g^−1^. This is also a direct consequence of the higher R values of the BCMg and BCF samples. Due to the short half-life of ^99m^Tc, the contact time of 240 min was chosen as the maximum possible for further batch experiment described below.

#### 3.2.4. Freundlich Isotherm

Freundlich adsorption isotherm describes the reversible and non-linear adsorption process. It is not restricted to monolayer; the multilayer formation is possible. Experimental data were obtained by the batch method. Freundlich isotherm was chosen as the most suitable for biochar samples. This is indicated by the values of the coefficients of determination, which are larger for the Freundlich isotherm. Freundlich: R^2^(BCS) = 0.85487, R^2^(BCMg) = 0.88979, R^2^(BCFe) = 0.98994; Langmuir: R^2^(BCS) = 0.74133, R^2^(BCMg) = 0.86398, R^2^(BCFe) = 0.98570. The result as a dependence of sorption capacity q_F_ on c_eq_ is in Figure 4. 

Freundlich constants k_F_, regression constants n_F_, and coefficients of determination R^2^ are shown in Table 8. The values of n_F_ are close to one for samples BCS and BCFe, greater than one for BCMg. Therefore, the adsorption is favorable in all cases, which corresponds to the concave shape of isotherms (see Figure 4). However, the best properties for the sorption have the samples BCFe and BCMg. The closest R^2^ value to 1 can be observed for sample BCFe, which means the most reproducible and credible data for calculation.

### 3.3. Dynamic Sorption Experiments

The course of the breakthrough curves in Figure 5a,c,e showed that, that relative count rate did not reach 100%, i.e., the count rate at the column output did not reach the value at the input to the column. There are more possible reasons of these results, namely, slow sorption kinetics, low residence time of the solution in bed, high dispersion into the bed connected with so called wall effect. As for the sorption kinetics, it was followed in Section 3.2.2 and, evidently, sample BCS has the sorption rate most slow (it has to be added, that this sample has the lowest sorption capacity, as well, see Figure 4a). The high dispersion is proved by the low values of P_e_, where the small height of the layer in the columns and probably also the uneven deposition of sorbent grains in the layer also contribute to this, including the already mentioned wall effect. 

A sample containing MgCl_2_ was expected to form a slightly acidic solution because MgCl_2_ hydrolyzed significantly. Therefore, it would be expected that the pH 1 of sorption solution would result in a rapid decrease in the pH of the fractions at the outlet of the column. However, it is just the opposite (Figure 5c). The pH rises very fast and is already above 9.5 in 2 BV. Thus, there must be another mechanism by which H^+^ ions are consumed from the sorption solution Now it is only possible to estimate the cause of this increase in pH, it is probably due to the capture of H^+^, or more precisely OH_3_^+^, on the surface groups of the layer sites of given sorbent (see determination of titration curves and Equation (3)). This is especially true for the sample BCMg. It is certainly interesting that in the case of BCS and BCFe samples, the pH value is set to four, probably due to the buffering efficiency of sites of these samples.

Evidently, a transport model based on a complementary error function, including the effect of dispersion in the layer expressed by the value of P_e_, can be applied to the modelling of experimental results. This is indicated by the blue dots (model) versus the black dots (experiment) in Figure 5b,d,f. The suitability of the model is proved by the WSOS/DF criterion (Equations (12)–(14)) which satisfies the condition that its value is ≤20 (Table 9). It follows that the model used corresponds to the experimental conditions. The values of the Peclet number P_e_, as was mentioned above, characterize the flow and is one of the parameters influencing the shape of the breakthrough curve. The values of sorption retardation coefficients R_s_ (Figure 6) are not constant, because the shapes of equilibrium isotherms are nonlinear, or, because its first derivative in point “c” is function of concentration (see Equations (46) and (47)). In other case, if equilibrium isotherm is linear, retardation coefficient is constant.

## 4. Conclusions

This work evaluates the use of biochar, its Mg and Fe engineered forms in relation to the ^99m^TcO_4_^−^/ReO_4_^−^ separation on these materials. The investigation of liquid phase pH in the Tc(VII) and Re(VII) sorption process has shown that the sorption preferably takes place in the acidic pH range. The sorption equilibrium is reached in two hours, *R* (sorption capacity) value reaches approximately the maximum for the BCMg and BCFe sample, for the BCS sample it reaches their capacity of 78%. This shows the suitability of the BCS sample treatment.

Breakthrough curves did not reach 100% in all cases. The reason, very probably, is that the sorption is slow due to the slow kinetics and higher dispersion of the liquid phase during the flow in the column, which is evident from the relatively low values of the P_e_ criterion. In addition, due to the short half-life ^99m^Tc, the low flow rate of the sorption solution and the small volume of the fractions obtained, it was not possible to perform an experiment in these cases up to 100% breakthrough. Furthermore, it was found that increasing the flow of the liquid phase does not contribute to the acceleration of ^99m^TcO_4_^−^/ReO_4_^−^ sorption.

Potentiometric titrations of the biochar samples were performed to find the types of surface sites and to determine the surface charge as a function of pH. At the same time, it was found that chemical (CEM) and ion exchange (IExM) models are best suited for the description of surface groups in a given case. The charge that exists on the surface of the samples during the sorption of ^99m^TcO_4_^−^/ReO_4_^−^ under the given conditions is positive and, as a result, the sorption of anionic forms of Tc(VII) and Re(VII) takes place. Namely, the sorption proceeds by ion exchange on protonated functional groups of the ≡SOH_2_^+^ type.

## Figures and Tables

**Figure 1 materials-14-00486-f001:**
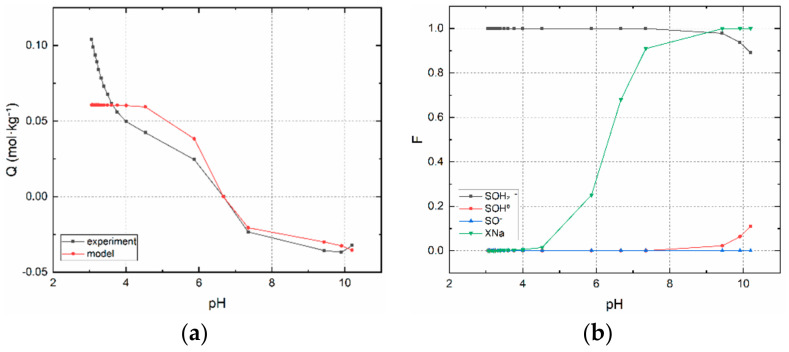
BCS experimental and calculated titration curves: (**a**) surface charge, Q = f(pH), (**b**) mol fractions, F = f(pH), of edge (SOH_2_^+^, SOH^0^, SO^−^) and layer (XNa) sites.

**Figure 2 materials-14-00486-f002:**
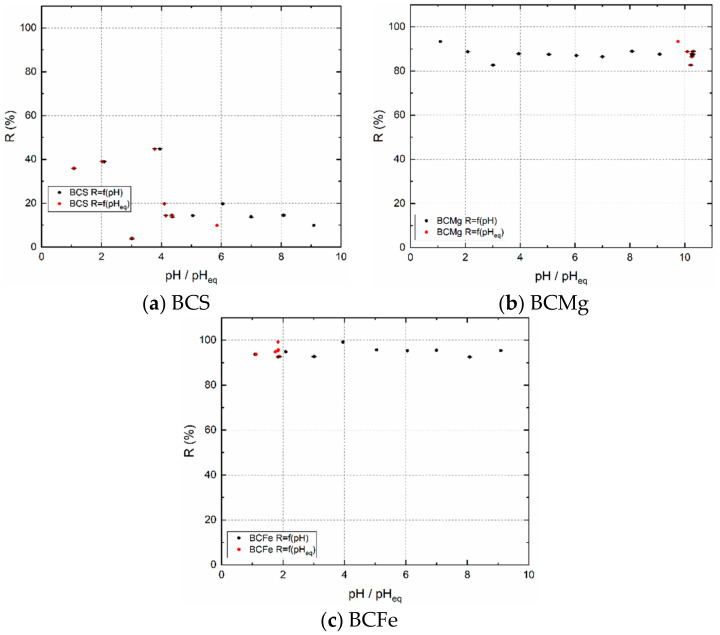
(**a**) BCS, (**b**) BCMg, (**c**) BCFe. The dependence of R on pH and pH_eq_. Tracer ^99m^TcO_4_^−^, m = 0.02 g, V = 2 cm^3^, t = 24 h.

**Figure 3 materials-14-00486-f003:**
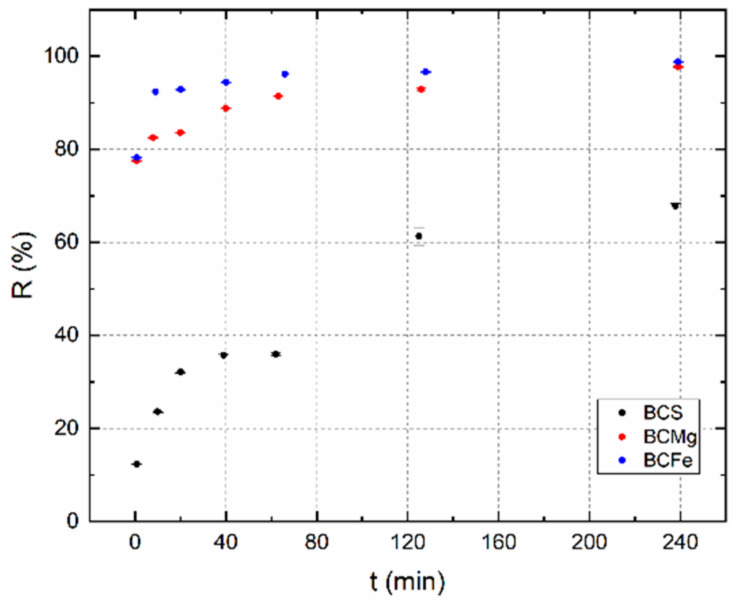
The dependence of R on contact time t. Tracer ^99m^TcO_4_^−^, m = 0.02 g, V = 2 cm^3^, pH = 1 (BCMg), pH = 4 (BCS & BCFe).

**Figure 4 materials-14-00486-f004:**
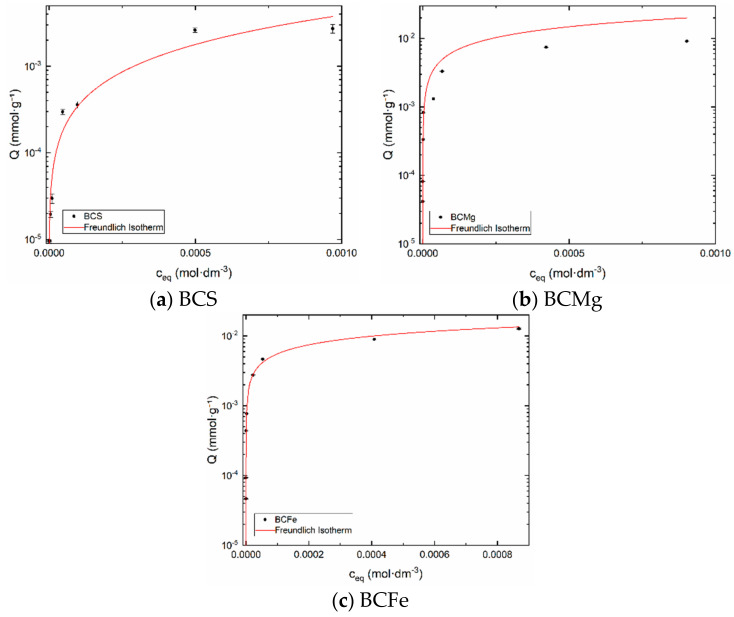
Freundlich isotherms. (**a**) BCS, (**b**) BCMg, (**c**) BCFe.

**Figure 5 materials-14-00486-f005:**
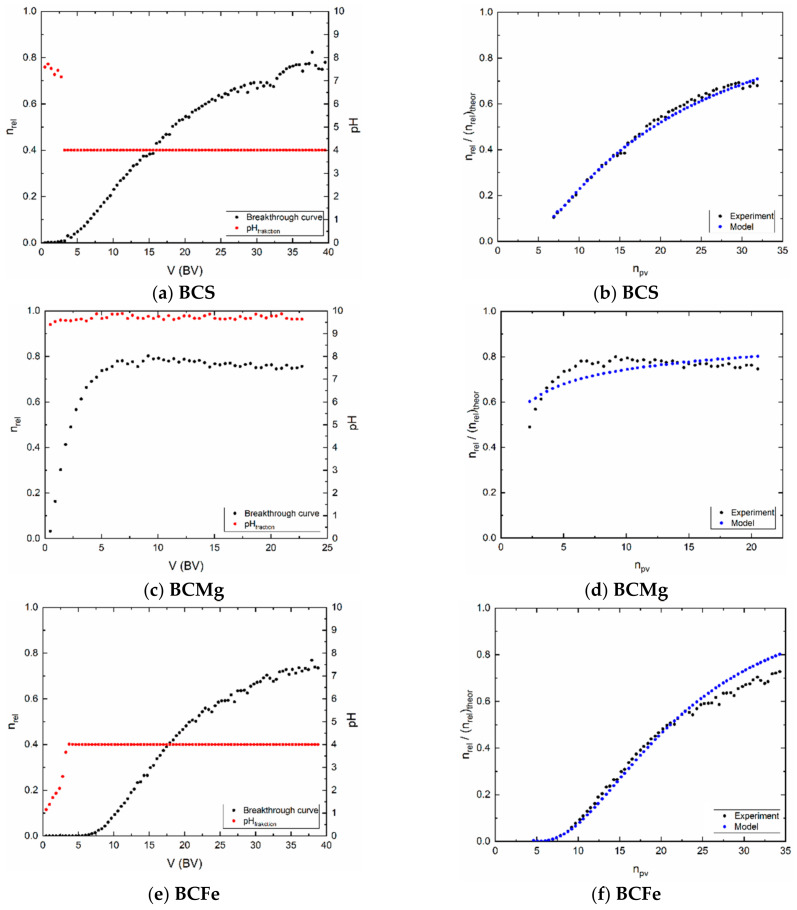
Left (**a**,**c**,**e**): Experimental breakthrough curves and actual pH of each fraction; Right (**b**,**d**,**f**): experimental breakthrough and the model.

**Figure 6 materials-14-00486-f006:**
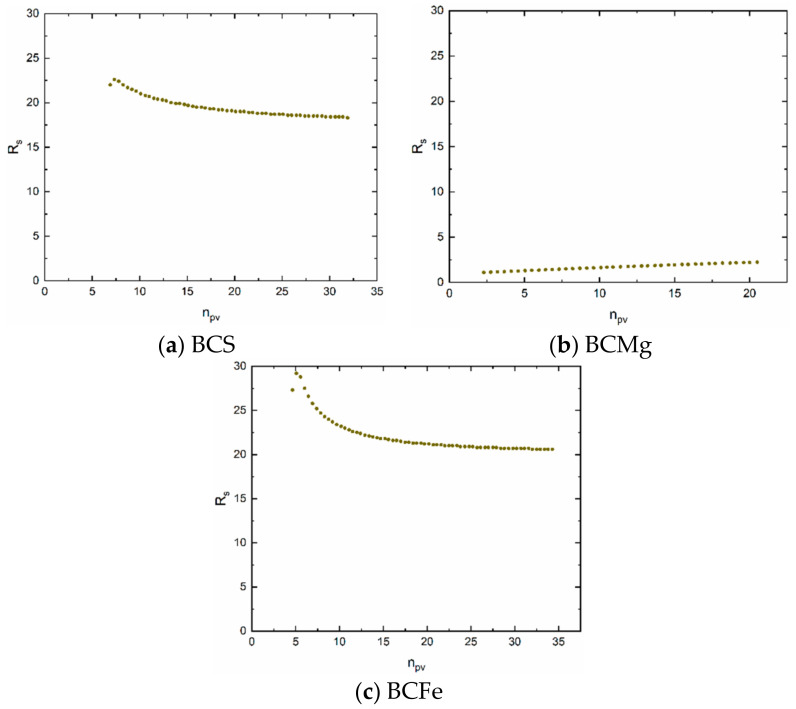
Retardation coefficients. (**a**) BCS, (**b**) BCMg, (**c**) BCFe.

**Table 1 materials-14-00486-t001:** Composition of sorption solutions and column bed masses.

Column		Sorption Solution	
Sample	*m* (g)	pH	*c*(NH_4_ReO_4_) (mol·dm^−3^)
BCS	0.3437	4	1 × 10^−3^
BCMg	0.4358	1	5 × 10^−^⁴
BCFe	0.5769	4	1 × 10^−3^

**Table 2 materials-14-00486-t002:** Identified elements by X-ray fluorescence (XRF) (K_α_ levels).

Sample	Element														
BCS			Si		P	Cl				Fe	Ni				Ag
BCMg	Mg		Si	S	P	Cl	K	Ca	Cr	Fe	Ni	Zn	Rb	Rb	Ag
BCFe		Al				Cl	K	Ca	Cr	Fe	Ni	Zn	Rb	Rb	Ag

**Table 3 materials-14-00486-t003:** Identified elements by energy-dispersive X-ray (EDX).

Sample	Elements								
BCS	C	O	Mg	Al	P	Cl	K	Ca	
BCMg	C	O	Mg	Al		Cl			
BCFe	C	O	Mg	Al	P	Cl	K		Fe

**Table 4 materials-14-00486-t004:** Calculated model parameters for sample BCS.

*K_1_*	*K_2_*	*K_ex_*	[SOH]_tot_ (mol·kg^−1^)
(1.54 ± 24.9) × 10^15^	(1.53 ± 0.16) × 10^11^	(8.71 ± 13.8) × 10^−1^	(6.07 ± 0.03) × 10^−2^
***WSOS/DF***	***χ^2^***	***σ_i_***	**[S]_tot_** **(mol·kg^−1^)**
10.3	1.34 × 10^2^	0.1	(8.94 ± 0.05) × 10^−2^

**Table 5 materials-14-00486-t005:** Mole fractions calculation result. (The values of F2 correspond to the values of pH_0_ and pH_end_, respectively. The values of F1 and F3 are function of ΣRe (=KReO_4_)).

ΣTc (mol·dm^−3^)	ΣRe (=KReO_4_) (mol·dm^−3^)	pH_0_	pH_end_	F1 [ReO_4_^−^]	F2 [HReO_4_]	F3 [KReO_4_]
10^−10^	10^−2^	0	3.6	0.34	9.80 × 10^−11^–3.90 × 10^−7^	0.66
10^−10^	10^−3^	0	3.6	0.84	2.40 × 10^−10^–0.96 × 10^−7^	0.16
10^−10^	10^−4^	0	3.6	0.98	2.28 × 10^−10^–1.13 × 10^−6^	0.02
10^−10^	10^−5^	0	3.6	0.99	2.88 × 10^−10^–1.15 × 10^−6^	0.002
10^−10^	10^−4^	0	10	0.98	7.11 × 10^−17^–1.13 × 10^−6^	0.02
10^−12^	10^−3^	0	3.6	0.84	2.41 × 10^−10^–9.60 × 10^−7^	0.16
10^−12^	10^−4^	0	3.6	0.98	2.83 × 10^−10^–1.43 × 10^−6^	0.02

**Table 6 materials-14-00486-t006:** Results of Eh(V) and pH measurements in NH_4_ReO_4_ solutions labelled with [^99m^Tc]NaTcO_4_.

c(NH_4_ReO_4_) (mol·dm^−3^)	Eh(mV)	pH
5 × 10^−5^	270.8	2
5 × 10^−4^	321.0	1
5 × 10^−4^	269.5	2
5 × 10^−4^	265.4	2
1 × 10^−4^	269.3	2
1 × 10^−4^	269.9	2
1 × 10^−3^	270.7	2
1 × 10^−2^	270.5	2
Zobell’s solution 1	−98.9	8.4
Zobell’s solution 2	−99.7	8.4

**Table 7 materials-14-00486-t007:** Minimum and maximum D_g_ values at which R reaches maximum.

Sample	pH	*R_max_* (%)	*D_g_* (cm^3^·g^−1^)	*D_g,min_* (cm^3^·g^−1^)	*D_g,max_* (cm^3^·g^−1^)
BCS	4	44.7 ± 0.2	80 ± 1	58	51,502
BCMg	1	93.3 ± 0.1	1383 ± 8	57	76,680
BCFe	4	99.2 ± 0.1	12,268 ± 284	58	52,015

**Table 8 materials-14-00486-t008:** Parameters of the Freundlich fit.

Scheme	*k_F_*	*n_F_*	*R^2^*
BCS	(1.95 ± 0.17) × 10^1^	(8.71 ± 0.14) × 10^−1^	0.85487
BCMg	(6.24 ± 2.08) × 10^28^	(965 ± 3.31) × 10^−2^	0.86398
BCFe	(215 ± 1.16) × 10^−1^	(946 ± 0.67) × 10^−3^	0.98994

**Table 9 materials-14-00486-t009:** Calculated parameters and fit criterion.

	*WSOS/DF*	*χ^2^*	*σ_i_*	*P_e_*
BCS	0.301	16	0.1	(1.93 ± 0.06) × 10^0^
BCMg	0.627	23.8	0.1	(19.5 ± 1.72) × 10^-2^
BCFe	1.16	72.9	0.1	(544 ± 4.68) × 10^-2^

## Data Availability

The data presented in this study are available on request from the corresponding author.

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
