# Peer review of "Pertechnetate/Perrhenate Surface Complexation on Bamboo Engineered Biochar"

_materials, 2021, doi:10.3390/ma14030486_

Round 1

Reviewer 1 Report

General Comments:

The paper can be interesting for specialists. The study presents a good idea of evaluation of biochar samples prepared from Acidosasa Edulis 13 bamboo.

The subject matter is within the scope of the journal. The manuscript adheres to the journal's standards.

The objectives of the study are well highlighted.

The key words permit found article in the current registers or indexes.

However, it must be revised in the light of following comments:

  1. Novelty of the work should be emphasized in the introduction part.
  2. Please check the numbering of the subchapters.
  1. In my opinion, the figures don’t have a good quality (the resolution is too short). I recommend to improve the quality of the figures.
  2. In the Abstract section it is mentioned that the ‘’equilibrium isotherms’’ were investigated, but in the manuscript only Freundlich isotherm it is presented. Even that this type of isotherm was chosen as the most suitable for biochar samples, it would be recommended to show the results of R2 for at least another isotherm, for example Langmuir model.
  3. Line 572-575--Please check the font
  4. Please verify the following references. Some typing mistakes were found.
  • Reference 2: The name of the Journal and the volume number must be written with Italic font;
  • Reference 16: Remove ‘’pp’’ before page range;
  • Reference 24: The name of the Journal and the volume number must be written with Italic font;
  • References 25, 26: The volume number must be written with Italic font.

Author Response

Dear reviewer,

thank you very much for your comments and suggestions. It is pleasure for us and we really appreciate the time you spend with reviewing of our manuscript. Please, find the answers in submitted file.

The authors.

Reviewer 2 Report

P2, L59: A term: Engineered/designer sorbent change into Engineered/designed sorbent

I suppose that the instumentation is old, so the ex-country Czechoslovakia (P6, L227; P8, L309; P8, L350) is listed instead of Czech Republic.

P9, L357: A shift has taken place!

P10, L424: A shift has taken place!

P10, L434: Identified elements by XRF are presented in Table 2, while elements identified by EDX are just listed.  I suggest the same presentation as well explanation of the differences.

P12, L473-476: A shift has taken place!

P14, L551: Figrue 2a change to Figure 2a

P16, L594: The scale on the y axis is not the same for Figures 4a, 4b and 4c

Author Response

(The authors gave the same response as above.)

Reviewer 3 Report

Include the word Bamboo in the title, much more commonly used than Acidosasa Edulis in activated carbon field.

I miss some explanation on relationship between the properties of the biochar, (surface area, pore volume, particle size, etc) and the adsorption behavior.

Table 1 could be completed adding the wt% of each element. If elemental analysis of C, H, S and O is combined with XRF the mass % can be estimated. Why in sample BCFe, Si, S and P are not detected? Are these elements removed during biochar preparations? Could be tested in the liquid of washing biochar?

Detail the main properties of biochar in this paper, as surface area, pore volume, micro or meso pore. These properties are used to explain why BCS reach only 78% sorption equilibrium while BCMg and BCFe reach 100%. If the last two have a larger surface area than BCS, the sorption equilibrium should be reached at longer times, so something else must influence, large pore diameter, shorter pores, smaller particle size, etc. This must be address on the manuscript.

Some surface groups on biochar can be identified with FTIR.

Did Mg and Fe leach during sorption experiments?

Why are not shown tritation curves of BCFE and BCMg in Figure 1?

Recommendation of give the adsorption conditions on the label of Figure 2 and 3. It makes easier for the reader to find data.

In figure 3, for the sample BCS more points are needed in times between 40 and 120 min

Author Response

(The authors gave the same response as above.)
